# Molecular Mechanisms of Muscle Fatigue

**DOI:** 10.3390/ijms222111587

**Published:** 2021-10-27

**Authors:** Dumitru Constantin-Teodosiu, Despina Constantin

**Affiliations:** Division of Physiology, Pharmacology and Neuroscience, School of Life Sciences, Queen’s Medical Centre, University of Nottingham Medical School, Nottingham NG7 2UH, UK; despina.constantin@nottingham.ac.uk

**Keywords:** skeletal muscle, atrophy, muscle function, fatigue

## Abstract

Muscle fatigue (MF) declines the capacity of muscles to complete a task over time at a constant load. MF is usually short-lasting, reversible, and is experienced as a feeling of tiredness or lack of energy. The leading causes of short-lasting fatigue are related to overtraining, undertraining/deconditioning, or physical injury. Conversely, MF can be persistent and more serious when associated with pathological states or following chronic exposure to certain medication or toxic composites. In conjunction with chronic fatigue, the muscle feels floppy, and the force generated by muscles is always low, causing the individual to feel frail constantly. The leading cause underpinning the development of chronic fatigue is related to muscle wasting mediated by aging, immobilization, insulin resistance (through high-fat dietary intake or pharmacologically mediated Peroxisome Proliferator-Activated Receptor (PPAR) agonism), diseases associated with systemic inflammation (arthritis, sepsis, infections, trauma, cardiovascular and respiratory disorders (heart failure, chronic obstructive pulmonary disease (COPD))), chronic kidney failure, muscle dystrophies, muscle myopathies, multiple sclerosis, and, more recently, coronavirus disease 2019 (COVID-19). The primary outcome of displaying chronic muscle fatigue is a poor quality of life. This type of fatigue represents a significant daily challenge for those affected and for the national health authorities through the financial burden attached to patient support. Although the origin of chronic fatigue is multifactorial, the MF in illness conditions is intrinsically linked to the occurrence of muscle loss. The sequence of events leading to chronic fatigue can be schematically denoted as: trigger (genetic or pathological) -> molecular outcome within the muscle cell -> muscle wasting -> loss of muscle function -> occurrence of chronic muscle fatigue. The present review will only highlight and discuss current knowledge on the molecular mechanisms that contribute to the upregulation of muscle wasting, thereby helping us understand how we could prevent or treat this debilitating condition.

## 1. Introduction: Muscle Architecture

The musculoskeletal system is one of the central organ systems in the body. It consists of muscles, tendons, cartilage, ligaments, connective tissues, and nerves. Muscle mass makes up about 40% of the whole-body weight. Muscles are vital to human life as they enable our anatomic integrity, maintain posture and balance, facilitate our daily routine with physical tasks, and help us execute our daily chores [1]. They also have essential roles in respiratory mechanics, offer protection of internal organs, and store minerals, fat, and carbohydrates in the form of glycogen. They facilitate the movement of substances inside the body and the generation of body heat. Finally, they help also return venous blood from the lower limbs to the right side of the heart.

There are three main types of muscle tissue: skeletal, cardiac, and smooth [2]. Skeletal muscles are fibrous tissues found in humans or animals mainly attached by tendons to the skeleton’s bones. They can contract/shorten upon neuro-mediated calcium stimulation, thereby moving the whole body while maintaining the position of parts of the body.

A skeletal muscle is made up of multiple fascicles, and each one includes numerous muscle fibers (Figure 1, [3]). The muscle fibers are, in turn, composed of myofibrils. The myofibrils are composed of overlapping, protein-made, thick (myosin) and thin (actin) myofilaments highly organized as sarcomere units, which are de facto the contractile units of the muscle. The sheaths made of connective tissue that encapsulate the bundle of myofibrils, muscle fibers, and the outer side of the muscle are named endomysium, perimysium, and epimysium, respectively.

Skeletal muscles contract via electric stimuli originating from the central nervous system (CNS). The impulses travel along nerves called motor neurons that insert into the muscle cells and branch, along with the blood vessels, into the epimysium and perimysium. The axons of the neurons then fork through the perimysium and enter the endomysium to innervate individual nerve fibers. The transfer of electrical signal from the motor neuron to the muscle fiber, which makes the latter contract, is facilitated by neuromuscular junctions. They are chemical synapses between a motor neuron and several muscle fibers, like those between regular neurons. Following stimulation by a nerve impulse, the terminal nerve releases the chemical neurotransmitter acetylcholine from synaptic vesicles. Acetylcholine then binds to nicotinic receptors located on an area of muscle fiber called motor endplate that entails folded sarcolemma. This binding opens the nicotinic receptor channels, and sodium ions flow into the fiber, depolarizing the muscle fiber membrane. The action potential generated spreads along the entire membrane to initiate excitation–contraction coupling. Propagation of the action potential is coupled to the release from the sarcoplasmic reticulum of calcium ions needed for contraction.

It is important to recall that not all skeletal muscle fibers are the same. There are broadly two types of skeletal muscle fibers, slow-twitch or type I and fast-twitch or type II muscle fibers [2]. Slow-twitch muscle fibers are resilient and mobilized for sustained, prolonged submaximal (aerobic) exercise and postural control. They contain numerous mitochondria and myoglobin and are highly aerobic compared to fast-twitch fibers. They are abundantly supplied with blood since more capillaries surround them. Fast-twitch muscle fibers generate more tension and more powerful forces, but for shorter durations, and fatigue rapidly. They are more anaerobic with less blood supply. The classification of muscle fibre types can be achieved by probing fibre type-specific molecular markers, such as myosin heavy chain isoforms. Typically, probing can be done through histochemical measurement of the myosin ATPase activity (low or high) or immunochemical identification with antibodies against myosin heavy chains [4].

Although the skeletal muscles contain both types of fibers, the ratios between fibers can differ depending on various factors, including muscle function, gene inheritance, age, and training. Most young and early adult humans in the general population have close to an even distribution of these two muscle fiber types in the muscles used for movement. However, the soleus muscle located on the back of the lower limbs and posterior back muscles involved in maintaining posture contain mainly slow-twitch muscle fibers. Conversely, the vastus lateralis, which is the primary muscle of the thigh, contains fast-twitch fibers predominantly. A further departure from the normal muscle fiber distribution can be found in the muscles of marathoners and elite cyclists, where a high percentage of slow-twitch muscle fibers can be identified. Contrary to this, the muscles of weightlifters and sprint runners show a high portion of fast-twitch muscle fibers. Although the ability of fibre types to shift from slow to fast and the other way around has been an ongoing subject of controversy, recent evidence may add more weight to the likelihood that such an event could occur with specific training regimens [5]. However, these findings can be easily concealed by the existence of significant inter-individual responses [6].

## 2. Muscle Fatigue

Muscle fatigue is broadly described as the inability of muscles to maintain the required force for a given task or generate an expected power [7,8]. In the case of multiple sclerosis, fatigue is viewed as a reversible motor and cognitive impairment, with reduced impetus and need to rest. It can appear suddenly or be triggered by mental or physical activity, dampness, acute infection, and even food ingestion [9].

Muscle fatigue can be classified as:
(1)*temporary* due to strenuous physical activities and is caused by accumulation in the intracellular space of working muscles with intermediary energy metabolism waste (e.g., lactate) or depletion of their energy-rich compounds (e.g., muscle glycogen store). The time to recover from muscle fatigue will depend on the extent of the intensity and length of the physical task. On average, the individual should be fully recovered within 3 to 5 days. The usual intervention to speed up muscle recovery involves massage, cold compression, and light analgesics’ intake. However, muscle fatigue lasting beyond 2 weeks should require medical attention.(2)*chronic* due to either:
(*i*)muscle atrophies (muscles waste away) due to immobilization, also called disuse atrophy, the presence of chronic inflammation in cardiovascular and respiratory disorders (e.g., heart failure, chronic obstructive pulmonary disease (COPD)), trauma, critical illness, medication (PPAR agonism),(*ii*)muscle atrophy with aging (sarcopenia), or(*iii*)neurogenic muscle atrophy due to obstructions or interference with different stages of nerve signal propagation from CNS to motor neuron plate due to disease or spinal injury. Depending upon location, it can be divided into central and peripheral [10]. Central fatigue is initiated at the central nervous system (CNS), such as in multiple sclerosis, thereby decreasing the neural drive to the muscle [11,12]. In contrast, peripheral fatigue is generated by changes at or distal to the neuromuscular junction such as in (a) autoimmune diseases caused by abnormal autoimmune reactions targeting neuromuscular synaptic proteins, as in Graves disease, Guillain-Barré syndrome, and myasthenia gravis, or (b) muscular dystrophies (MDs) due to genetic defects. MDs are progressive and debilitating diseases characterized by muscle wasting and progressive weakness. Duchenne (lacks the dystrophin component of the dystrophin-glycoprotein complex), Becker (contains a mutated dystrophin gene), and Limb-girdle type IIA (includes a mutation in the gene coding for calpain 3-P94) are typical examples of MDs.

The overall abnormally chronic fatigue impinges severely on the functional status and quality of life of the individuals affected by restricting their habitual daily activities and reduced survival in some circumstances like those described earlier at point (*iii*).

## 3. Reduction of Muscle Mass and Function

Individuals that preserve healthy skeletal muscles over their life span have a better quality of life as good muscle function empowers them with a well-desired physical autonomy later in life. Skeletal muscle wasting begins around the early 30 s, and muscle atrophy is associated with several severe morbidities and mortalities. The presence of progressive skeletal muscle wasting affects muscle structure severely, thereby compromising muscle function. This is intimately linked to increased risk factors and the prevalence of morbidity and mortality. Thus, reducing skeletal muscle mass due to disease-mediated waste or inactivity is an independent predictor of survival during cancer and old age [13,14,15,16]. Yet, some may argue that, although the strength of skeletal muscle largely depends on the mass of this tissue, the low muscle mass does not always explain the strong association of muscle strength with mortality, which may indicate that muscle strength as a marker of muscle quality may be more significant than quantity in assessing mortality risk [17,18]. In quantitative terms, peripheral muscle wasting during disease can account for up to 2% loss of muscle mass per day or up to 10% loss per week [19], which delays patient recovery and rehabilitation.

The main underlying factor behind the loss of muscle mass, malnutrition, and negative nitrogen balance is an increase in skeletal muscle protein degradation. This occurs on a background of inflammatory responses to trauma or infection, increased circulating cytokine, glucagon, epinephrine, and glucocorticoid treatment, hyperglycemia-mediated secondary infections, and induction of muscle insulin resistance. The immobilization is also an important factor triggering the preferential myosin loss, atrophy, and loss of specific force in fast- and slow-twitch muscle fibers with the loss of strength, especially in the quadriceps and extensors [20]. The critical implications of muscle protein loss extend to poor clinical outcomes such as wound healing, decreased ambulation, and increased risk of thromboembolic complications. There is also evidence that trauma and sepsis can lead to pulmonary complications due to a catabolic response in the respiratory muscles [21], extending to peripheral skeletal muscles [22].

Although the amount of protein that is degraded in healthy subjects of a given age equals typically the amount of protein synthesized, the whole-body protein turnover (protein synthesis + protein degradation) decreases gradually with ageing after peaking through puberty. Past 50 years of age, individuals lose approximately 1–2% of muscle mass per year due to reduced fiber number and atrophy of the remaining fibers. By 80 years, humans generally lose 30–40% of skeletal muscle fibers. A quarter of people under 70 and 40% under 80 show signs of sarcopenia [23]. Of note, males are more susceptible than females to become sarcopenic with ageing [24].

## 4. Major Molecular Mechanisms of Underlying Muscle Wasting

Skeletal muscle proteolysis is regulated by at least four metabolic pathways: (1) ubiquitin ATP-dependent proteasome (UPP), (2) autophagy lysosomal, (3) calcium-dependent, and (4) myostatin mediation. Animal models have previously shown that most, but not all, cellular proteins are degraded by the proteasome pathway [25]. Ubiquitinated proteins are then recognized, unfolded, and degraded by the multicatalytic 26S protease (proteasome) complex (Figure 2). The action of several enzymes achieves ubiquitination: ubiquitin-activating (E1) enzyme, ubiquitin-conjugating (E2) enzymes, and ubiquitin ligases (E3). Two muscle-specific ubiquitin ligases, muscle atrophy F-box (MAFbx) and muscle RING finger 1 (MuRF1), have emerged as critical regulators of skeletal muscle proteolysis under catabolic conditions [26]. In line with the above, the finding of increased concentrations of ubiquitinated proteins in muscle homogenates from myopathic septic patients [27] and a several-fold increase in the levels of 20S proteasome (the catalytic core of the 26S proteasome complex) mRNA and protein expression in seriously ill patients that need immediate life support [28,29] would highlight the magnitude of protein breakdown in critical illness. The overall elevated proteolysis rate can also be clinically identified from increases in urinary 3-methyl histidine excretion, an index of myofibril protein breakdown [30].

Concurrent with the increased expression of the 20S, the muscle-specific ubiquitin ligases, MAFbx and MuRF1, are also up-regulated in critical illness [28,31], suggesting that an upstream control of the components of the ubiquitin-proteasome pathway exists (Figure 2). Several cytokines that increase oxidative stress have received attention as potential underlying triggers to up-regulation of the UPP [32]. Although they are not exclusive, the most cited cytokines are tumor necrosis factor-alpha (TNF-α) and interleukin 6 (IL-6). An initial increase in cytokines’ levels is usually beneficial since it triggers the host’s innate immune responses to protect against infections. However, in sepsis, which is a leading cause of death in intensive care units, the excessive production of these cytokines can reverse this positive effect of an inflammatory response into tissue injury and multiple-organ failure [33]. Indeed, a recent study provided evidence of remarkably high TNF-α and IL-6 expression levels in muscles of critically ill patients [28]. Such increases might be reasonably predictable to reduce insulin receptor substrate-1 (IRS-1) binding [34], thereby inhibiting Akt signaling [35] (Figure 2). Moreover, TNF-α is assumed to directly inhibit protein kinase B (Akt) protein [36].

The net result of both events would be de-phosphorylation (activation) of the forkhead (FOXO) family of transcription factors and upregulation of FOXO1 downstream target genes, including MAFbx and MuRF1 [37]. This would assign the FOXO1 transcription factor an important role in the development of muscle atrophy, although as will be described later, the responsibility of FOXO1 activity also extends to its involvement in the instauration of an insulin resistance state. In line with this, Constantin et al. [28] confirmed down-regulation of muscle Akt1 phosphorylation, which occurred concomitantly with increased MAFbx and MuRF1 mRNA and protein expression in critical illness. Concerning 20/26S proteasomes, the modulation of UPP by cytokines could also be exerted via activation of muscle pattern recognition receptors like Toll-like receptor 4 (TLR4) signaling. TLRs activate the nuclear factor kappa B (NF-KB) pathway, further exacerbating cytokine expression through several adapter molecules, including myeloid differentiation primary response 88 (MyD88) and adapter protein in responding to TLR activation (TRIF, Figure 2).

Bed rest is another cause for concern in critical illness regarding muscle mass loss [38]. These patients are often sedated and fully immobilized for days. The harmful effects of immobilization on muscle loss are fast (within hours, [39]). It results in rapid decreases in muscle mass, muscle cell diameter, and the number of muscle fibers [32]. Controlled studies (no inflammation/average cytokines) in human subjects investigating molecular changes underpinning the sole effect of 2-week cast leg immobilization revealed that the 5% decrease in quadriceps mass was linked, but not limited to, greater expression of the 20S proteasome and the muscle-specific proteolytic genes MAFbx and MuRF1 [40]. Bed rest over 48 hrs in healthy subjects also induced profound changes in blood flow and decreased perfusion pressure [32], exposing the muscle to ischemia and potentially contributing to oxidative stress induction. Indeed, during surgeries that require full stop blood flow to the operative leg, components of the catabolic FoxO3a (i.e., MuRF1, MAFbx, and Bnip3) pathway, as well as the cellular stress pathways (stress-activated protein kinase (SAPK)/JNK and mitogen-activated protein kinase (MAPK)) are up-regulated [41]. These collectively contribute to the activation of protein degradation.

## 5. The Role of Lysosomal Autophagy in Muscle Protein Breakdown

Autophagy is a highly conserved catabolic process in which lysosomes engulf cytoplasmic macromolecules and organelles for degradation. Although basal levels of autophagy support healthy maintenance of metabolic homeostasis through sustaining a balance between protein synthesis and degradation and organelle biogenesis and degradation, amplified autophagy due to metabolic and inflammatory diseases also contributes to the disruption of the protein synthesis–degradation cycle. Previous animal-based research suggested that in cachexia, lysosomal-mediated protein degradation can account for almost one-third of muscle protein loss [42]. These membrane-enclosed organelles contain several acidic pH 5-optimal proteases, including cathepsins B, H, and D, and many other hydrolases, for example, aspartate proteases and Zn^2+^ metalloproteases. Some cytosolic proteins and even cellular organelles are degraded in lysosomes after being engulfed in autophagic vacuoles that fuse with lysosomes (Figure 2). In line with animal studies, investigations in humans have confirmed that muscle cathepsin-D mRNA expression and enzyme activity increase in, for example, cachexia and trauma [43]. More recently, Constantin et al. [28] showed evidence of cathepsin-L up-regulation in a more heterogeneous group of patients with critical illness, pointing to cathepsin-L as an active muscle wasting component in critical illness.

## 6. Muscle Calpains’ Expression Is Increased in Muscle Wasting

Calpains are cytosolic Ca^2+^-activated, non-lysosomal, cysteine proteases, of which μ- and m-isoforms (also known as calpain-1 and calpain-2, respectively) are the most ubiquitously expressed, including in skeletal muscle. Calpain-1 and calpain-2 are believed to degrade cellular ‘cement’ proteins, such as titin, vinculin, talin, desmin, and troponin within the myofibrils’ architecture (Figure 2). The action of these proteases is a prerequisite for the UPP and lysosomal pathways to degrade the released actin and myosin and other ubiquitin-tagged cytoskeletal proteins. Since cytosolic Ca^2+^ concentrations are consistently elevated in sepsis, chronic inflammation, and ischemia [44], it would not be unreasonable to assume that calpain-1 and calpain-2 activation will also be a feature of critical illness. Indeed, increased expression and activity of calpain-1 and calpain-2 is well documented in human muscle wasting models [45].

## 7. Muscle Myostatin Expression Is Increased in Muscle Wasting

Myostatin or growth differentiation factor 8 (GDF8) is a member of the transforming growth factor (TGF)-α superfamily, which functions as a negative regulator of satellite cell proliferation and differentiation, muscle growth, and development (Figure 2). This is thought to occur through the myostatin-mediated inhibition of the myogenic regulators’ myoblast determination protein 1 (MyoD), muscle-specific basic helix-loop-helix (bHLH) transcription factor (myogenin), and myogenic factor 5 (Myf5) [46]. Myostatin is also important in the regulation of human muscle mass because it is involved in the control of both muscle protein synthesis by inhibiting anabolic signaling, which is translation initiation, through inhibiting the Akt/mTOR)/p70S6k signaling pathways [47] and muscle protein breakdown [48]. Consistent with this role, increased myostatin mRNA and protein expression appears to be a significant feature of human skeletal muscle wasting in multiple noncommunicable diseases [28].

## 8. Apoptosis Is Increased in Muscle Wasting

It also appears that activation of apoptotic signal transduction (programmed muscle cell death) during muscle denervation is another player involved in regulating denervation-induced muscle atrophy along the four major earlier-presented protein degradation pathways [49]. Although myonuclei’s apoptosis may contribute to the loss of muscle mass, the mechanisms underlying this process are still largely unknown. The study by Siu et al. [49] indicated that the apoptotic Bax and Bcl-2 proteins, both members of the bcl-2 family, and the mitochondria-associated apoptotic factors, including cytochrome c, the second mitochondria-derived activator of caspase (Smac/DIABLO), and the apoptosis-inducing factor (AIF), were all increased in denervated muscles. Moreover, denervation augmented the protein content of heat shock protein 70 kDa (HSP70). In contrast, the mitochondrial isoform of superoxide dismutase (MnSOD) protein content was reduced, which indicated that denervation might have induced cellular and/or oxidative stress.

Conversely, some argue that the skeletal muscle does not undergo apoptosis during either atrophy or programmed cell death, aka apoptosis, thereby supporting the theory that the nucleus persists once a muscle fiber has acquired it [50].

## 9. Critical Illness Is Associated with Muscle Insulin Resistance

Patients with critical illness develop insulin resistance, which refers to a normal physiological concentration of insulin producing a less than expected biological response. Present evidence suggests that the simultaneous up-regulation of the central muscle UPP-protein degradation (via muscle-specific ligases MuRF1and MAFbx) [37] and muscle insulin resistance [51] share, via Akt1, PPARδ, and the FOXO family of transcription factors, a common signaling pathway. It appears, therefore, that FOXO1 is a gatekeeper of a central crossroad between the most important proteolytic pathway (UPP) and insulin resistance. The involvement of the FOXO1 transcription factor in the etiology of muscle insulin resistance in critical illness is mediated via increased pyruvate dehydrogenase kinase 4 (PDK4) transcription [28]. Since PDK4 specifically phosphorylates (inactivates) pyruvate dehydrogenase complex (PDC), it can be therefore accepted as a rate-limiting factor in carbohydrate (CHO) oxidation [52]. Insulin, along with muscle contraction, is the most important physiological activator of PDC [52]. Although insulin is an effective treatment in increasing whole-body glucose disposal rate, its use confers an increased risk of hypoglycemia, which inadvertently increases mortality [53].

## 10. Muscle INSULIN Resistance Is Increased with PPAR Agonism and Statins’ Treatment

The activity of muscle mitochondrial pyruvate dehydrogenase complex (PDC), the enzyme that controls the rate of CHO oxidation, seems to be negatively affected by peroxisome proliferator-activated receptor (PPAR) agonists or statins medication [54,55]. We have recently shown that PPARδ agonism, used clinically to increase muscle fat oxidation, up-regulates PDK4 mRNA and protein expression in resting skeletal muscle via changes in the Akt1/FOXO/MAFbx and MuRF1 signaling pathway [54]. As PDK4 isoform is the most potent inhibitor of PDC, activation of the PDK4 would tilt the muscle fuel metabolism towards decreased CHO use and enhanced lipid utilization. This change in fuel use induced by PPARδ agonism is also paralleled by the initiation of an atrophy program [54]. Since CHO demand is markedly increased during muscle contraction, one would expect that PPARδ agonism would negatively affect muscle function during sustained contraction. Indeed, data collected from a group of three rodent muscles showed that during prolonged contraction, PPARδ agonism inhibited muscle CHO oxidation at the level of PDC, and this was paralleled by the activation of anaerobic metabolism, which collectively impaired contractile function, thereby inducing premature muscle fatigue [56].

Muscle fatigue mediated by medication is not restricted only to PPAR agonism. Thus, statins’ therapy used clinically for cholesterol reduction is also associated with myopathic changes through a poorly defined mechanism. Many patients taking statins often complain of muscle pain and weakness. Statin-related myopathy has varying degrees of severity, ranging from muscle myositis and myalgia (muscle aches or weaknesses with and without increased serum creatine kinase (CK) levels, respectively) and, in the acutest case, rhabdomyolysis (>10 times the upper limit of average serum CK levels) [57]. We previously reported in an in vivo model of statin myopathy that simvastatin supplementation down-regulated PI3k/Akt signaling, independently of RhoA, and up-regulated FOXO transcription factors and downstream gene targets known to be implicated in proteasomal- and lysosomal-mediated muscle proteolysis, CHO oxidation, oxidative stress, and inflammation [55]. These changes occurred before evidence of extensive myopathy or a decline in the muscle protein to DNA ratio.

One can note that jointly for both medication-mediated muscle waste [54,55], an activation of muscle PDK4 and downregulation of muscle PDC activity occurred, which is suggestive of the instauration of an insulin resistance state. Evidence suggests that dichloroacetate (DCA), an agent that inhibits PDK4, thereby activating muscle PDC, can rapidly improve whole-body glucose disposal without causing hypoglycemia in healthy volunteers [58]. Therefore, increasing CHO oxidation in vivo using DCA should reverse the activation of FOXO and PPARδ downstream targets and reduce statin myopathy. Indeed, when DCA was given with simvastatin, the body mass gain and food intake were maintained; myopathy was abrogated; muscle proteasome activity, PDK4 mRNA, protein, and MAFbx cathepsin-L mRNA lowered; and activity of PDC increased compared with simvastatin alone [59]. Equally important and relevant to the current topic, the administration of DCA has also been proven to protect in vivo rodent animal models against sepsis [60].

## 11. ROS Involvement in Muscle Wasting

Lengthy periods of inactivity, bed immobilization, and disease (e.g., cachexia) show decreases in both muscle contractile function and muscle fiber size. It is recognized that the inactivity-mediated changes in muscle fibers, although not limited to this, result from a concurrent increase in muscle protein degradation and a decrease in protein synthesis [61,62,63]. While many facts about the mechanism underlying protein degradation are known, the signaling pathways controlling muscle protein balance remain undetermined. A trigger factor often named is the excess production of reactive oxygen species (ROS). The main site of ROS production is in the cytoplasm. One of the most well-known sources of ROS is the nitric oxide (NOX) family NADPH oxidase enzymes, which are proteins that transfer electrons across cellular membranes [64]. At large, the electron acceptor is oxygen, and the product of the electron transfer reaction is superoxide. The biological function of NOX enzymes is, therefore, to generate reactive oxygen species. In addition to NOX-dependent ROS production, the nitric oxide synthases (endothelial [eNOS], neuronal [nNOS], and inducible [iNOS]) are also sources of ROS [65]. The observation that prolonged periods of contractile inactivity also led to increased production of reactive oxygen species (ROS) in muscle fibers suggests that ROS could be an important signaling molecule contributing to muscle atrophy. Theoretically, increased ROS can accelerate proteolysis and autophagy [66] and depress protein synthesis via (1) limiting the ability of the mechanistic target of rapamycin (mTOR) to phosphorylate the eukaryotic translation initiation factor 4E-binding protein 1 (4E-BP1), which is an inhibitor of the eukaryotic translation initiation factor 4F (eIF-4F) complex [62] or (2) the 5’ AMP-activated protein kinase (AMPK) pathway [65]. Yet, it is still controversial whether oxidants are a major contributor to disuse muscle atrophy. Therefore, it is prudent to say that, although ROS production in skeletal muscle is associated with muscle wasting, it remains much of a debate as to whether oxidative stress is a cause or consequence of muscle atrophy [67].

## 12. Protein Synthesis Is Inhibited in Critical Illness but Is Accompanied by the Initiation of an Anabolic Restoration Program

Changes in whole-body or lower limb protein synthesis have been documented in trauma and critical illness [68,69,70]. The existing evidence suggests that the loss of muscle mass is also a direct result of a decline in muscle protein synthesis arising from a reduction in translational efficiency rather than a fall in ribosome number [71]. Specifically, de-phosphorylation (inactivation) of Akt1 decreases protein translation initiation by dampening the mTOR signaling cascade, thereby inhibiting the ribosomal S6 kinase (p70s6k) and activating (1) 4E-BP1, a negative regulator of the translation initiation factor eIF-4E [72] and (2) glycogen synthase kinase 3β (GSK3β), which is another negative regulator of protein synthesis via inhibition of the eukaryotic translation initiation factor 2B (eIF-2B) [73] (Figure 2). Furthermore, the reduction of Akt phosphorylation in catabolic conditions opens the door to up-regulation of MAFbx and MuRF1 and, thereby, activation of muscle protein breakdown [37]. Although the initial proof of concept was tested in cell-based and animal models [60,74], Constantin et al. [28] recently documented decreases in Akt1, GSK3β, mTOR, p70s6k, and 4E-BP1 protein phosphorylation in patients with a variety of diseases (e.g., sepsis, aggravated systemic inflammation, organ failure, and trauma), which were associated by widespread up-regulation of molecular events controlling muscle protein breakdown. Moreover, these events were accompanied by the novel observation that the suppression of muscle anabolic signaling in this cohort of patients was paralleled by a comprehensive increase in mRNA expression of these same signaling proteins, pointing to the initiation of a cellular anabolic restoration program in critical illness. These findings bear similarity to the simple observation that muscle total RNA, rather than specific mRNAs [28], is significantly higher in critical illness than in age- and gender-matched controls [19,29].

## 13. Muscle Wasting with Ageing Sarcopenia

Sarcopenia is a progressive loss of muscle mass and function in the absence of any noticeable disease. It is frequently used to describe a collection of cellular signaling pathways’ responses, which, over time, contribute to the accumulation of damaged cells and a collection of outcomes such as decreased muscle strength, reduced mobility and function, increased fatigue, increased risk of metabolic disorders, and increased risk of falls and skeletal fractures [75]. Most importantly, sarcopenia can occur at any age because of disuse or malnutrition. In younger individuals, the loss of muscle mass is reversible, whereas, in older subjects, the muscle loss appears irrecoverable. With age, there is an increased susceptibility to contraction-induced injury and a decreased ability to recover from injury leading to muscle atrophy and weakness [76]. As the abundance and recruitment of satellite cells or muscle stem cells are low in the ageing myofibrils, skeletal muscle regeneration, growth, and maintenance are severely impaired [77]. Of note, the reduction in the number of satellite cells in type II fibers of atrophic muscle in elderly individuals is notably remarkable, and such a fiber type-specific reduction in satellite content could represent an important factor in the etiology of sarcopenia [78]. The age-related motor unit remodeling (muscle fibers are progressively denervated or reinnervated by compensating neurons) leads to muscle force and power loss, thereby adding further weight to enduring muscle fatigue [79]. Although with age, there is a general marked reduction (35%) in the cross-sectional area of muscle compared with younger individuals [80] along with a lower strength in all types of muscle fibers, the power in the muscles of men remains greater than that in women mainly because of the greater muscle mass that men possess, and, therefore, is less fatigable [81]. An increase in fat infiltration–lipotoxicity of muscle may be an additional factor contributing to progressive muscle fatigue in the elderly [82].

A diminished function of satellite cells at old age may also impede preservation and repair from contraction-induced injury and contribute to age-related muscle wasting. Satellite cell function may also be affected by circulating factors, as muscle regeneration in old mice sharing the serum of young mice was not impaired [83]. For instance, the presence of chronic, low-grade systemic inflammation in older subjects may be one of those factors. Indeed, the inflammatory cytokine TNF-α negatively affects the muscle regenerating capacity. TNF-α destabilizes the myoblast determination protein 1 (MyoD), a muscle-specific transcription factor involved in satellite cell proliferation and differentiation, and induces apoptosis of satellite cells, particularly at old age. It was also proposed that some of the other effects mediated by TNF-α involve the expression of inhibitors of cell differentiation proteins [84], such as chromatin-modifying proteins, which play an important role in muscle cell differentiation via interfaces with key muscle-promoting transcription factors such as MyoD and the myocyte enhancer factor-2 (MEF-2) family [85]. It may be argued that the increase in TNF-α during the normal inflammatory response helps rather than impairs. Still, it must be remembered that the negative effect of systemic inflammation on muscle strength at old age may only become apparent when it exceeds a certain threshold and persists for an extended period, such as a life span [84].

## 14. Muscle Wasting with Chronic Chemical Exposure

Further related to developing muscle fatigue following exposure to a chemical is Gulf War syndrome or Gulf War illness (GWI). This term is an umbrella term that covers chronic and multi-symptomatic disorders affecting returned military veterans of the 1990–1991 Persian Gulf War. Many acute and chronic symptoms have been linked, including musculoskeletal weakness, muscle pain, and fatigue. One of the suggested underlying mechanisms for GWI is a chemically induced impairment of liver function, with the spillage of stored vitamin A compounds (“retinoids”) into the circulation in toxic concentrations, resulting in a chronic endogenous form of hypervitaminosis A [86]. Using a rodent model, Ramirez-Sanchez et al. [87] provided evidence that animals exposed to chemicals like those envisaged to have been used in the battlefield showed marked muscle atrophy, decrease in myofiber area and limb strength, and reduced treadmill time/distance. Muscle wasting pathway proteins were upregulated, while those promoting growth, mitochondrial function, and muscle ATP levels decreased. Proteomic analysis of muscle also documented unique alterations in the mitochondrial and metabolic pathways. Thus, like the outcomes described following exposure to prescribed medication, chemicals related to GWI adversely impacted key metabolic pathways leading to muscle atrophy and loss of muscle function.

Similarly, heavy metals, such as lead and mercury, or poisoning through environmental contamination also show, among many symptoms, muscle wasting and muscle fatigue. Since the leading target of this type of poisoning is the nerve, it is not surprising that this type of poisoning outcome is clinically classified as peripheral motor neuropathy. Peripheral neuropathy from lead poisoning typically affects the distal upper limbs [88]. Lead neuropathy can be due to demyelination or impairment of axon function. Consequently, and like the injury of lower motor neurons, the impulse coming from the CNS is impaired or even severed, and flaccid paralysis and induction of muscle atrophy and reduction in muscle mass, fiber cross-sectional area, and strength are rapidly developed.

## 15. Muscle Fatigue Associated with Neurogenic Muscle Atrophy Induced by Viruses

A classic example of a disease associated with chronic muscle fatigue is multiple sclerosis. Here, the suppressor function of regulatory T cells (Tregs), which have a role in regulating or suppressing other cells of the immune system, thereby controlling the immune response to self and foreign particles (antigens), is impaired for vague reasons. We recently provided evidence that supports a novel mechanism underlying diminished Treg function in multiple sclerosis. Thus, infections that activate the toll-like receptor 2 (TLR2) in vivo (specifically through TLR1/2 heterodimers) could shift the ratio between the Treg and the HIV inhibitor T helper 17 (Th17) cells’ balance toward a pro-inflammatory state in multiple sclerosis, thereby promoting disease activity and progression [89].

We have also associated the expression of human endogenous retroviruses (HERVs) with multiple sclerosis [90]. The multiple sclerosis-related retroviruses (multiple sclerosis-associated retrovirus (MSRV)/Human Endogenous Retrovirus-W (HERV-W)) have the potential to activate inflammatory immunity, thereby promoting both susceptibility and progression towards multiple sclerosis. The observation that people infected with the human immunodeficiency virus (HIV) may have a lower risk of developing multiple sclerosis than those non-infected with HIV offers, indeed, support for a connection between HERVs and multiple sclerosis. This may be due to suppression of HERV expression by antiretroviral therapies (ART) used to treat HIV.

COVID-19 is a recent viral infection that has spread worldwide and has been identified to affect multiple organ systems, including the nervous system. There is now a mounting body of evidence to suggest the existence of a long-term, post-Covid muscle fatigue syndrome even after mild cases of viral infections. There is, so far, no evident description of the underlying pathology. Muscle deconditioning, immune- or virus-mediated neuropathy, and exercise hyperventilation have been hypothesized to play an essential role in developing debilitating symptoms [91]. Nevertheless, due to the presence of robust and durable systemic inflammatory responses to the viral load (such as the formation of ROS and NO by immune cells during chronic inflammation) in conjunction with a lengthy bed immobilization and medication initially intended to dampen the immune responses, which otherwise also stimulates muscle atrophy (e.g., the steroid dexamethasone), it may not be, therefore, surprising that these factors could contribute to accelerating the energy-independent proteolytic activation of protein degradation via calcium (calpains) and TNF-α (E3-ligases) along with the dexamethasone-induced upregulation of myostatin [92]. Collectively, these outcomes might explain the initiation of muscle atrophy and muscle fatigue.

## 16. Conclusions and Future Perspectives

In summary, chronic muscle fatigue primarily stems from the loss of contractile function secondary to skeletal muscle wasting induced by diverse insults that range from chronic inflammation, injury, prolonged immobilization, medication, or neurogenic origin. Given the heterogeny of triggers that initiate muscle wasting, it may be challenging to identify appropriate all-inclusive interventions. Prescribed immunosuppressants such as calcineurin inhibitors, interleukin inhibitors, selective immunosuppressants, and TNF-α could potentially alleviate the disease phenotype. There is, however, a clinical paradox as, in their pursuit to save lives, clinicians would prescribe to the patient medication that, as earlier revealed, induce muscle wasting *per se* (e.g., glucocorticoids, PPAR agonism). Therefore, it appears that quite often, a combination of robust and relentless muscle rehabilitating programs such as those involving resistive exercise, along with a protein-rich diet intervention may be a practical, straightforward approach to improving muscle mass and muscle function, which would thereby enhance the overall health and quality of life of the individuals affected. Sadly, those cases of a neurogenic origin would necessitate more advanced medical care, although the beneficial health return is momentarily minimal.

## Figures and Tables

**Figure 1 ijms-22-11587-f001:**
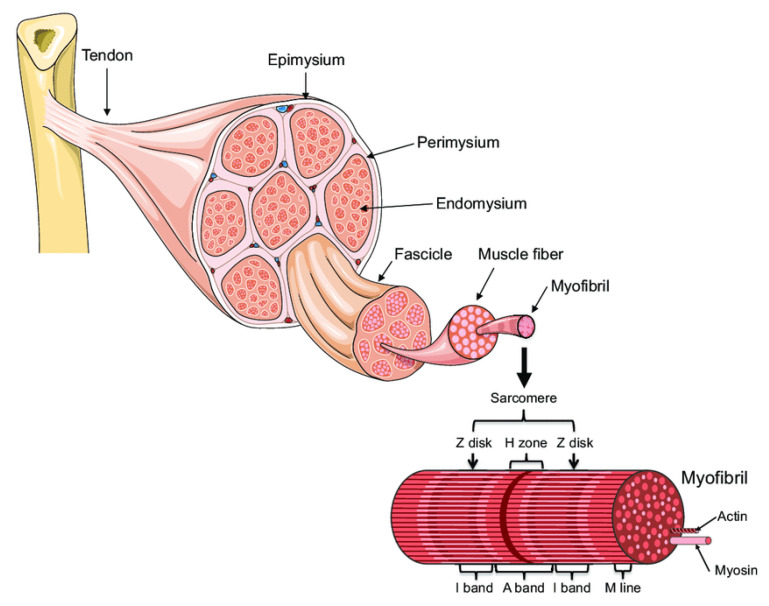
Muscle organization (A. Bonetta and LF Bonewald, originally adapted from Servier Medical Art—https://smart.servier.com/smart_image/tendon-anatomy/ accessed on 1 October 2021) [3].

**Figure 2 ijms-22-11587-f002:**
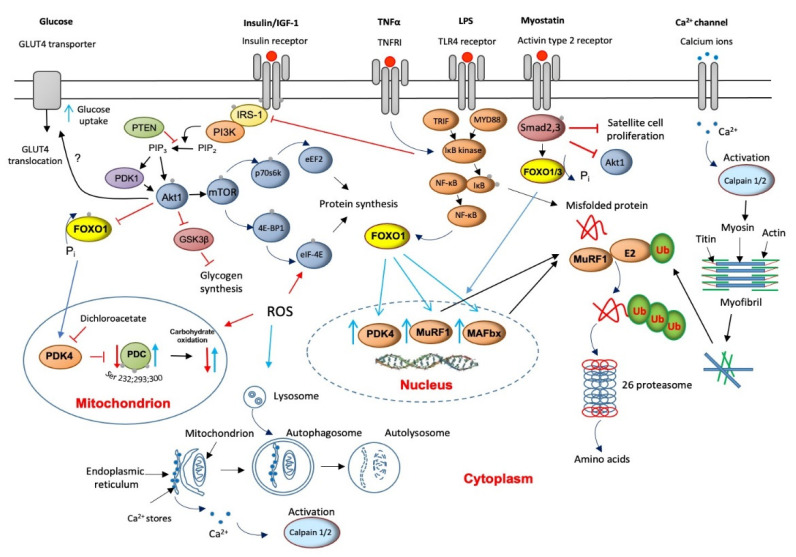
A schematic illustration of the main molecular mechanisms underlying muscle wasting and thereby to muscle fatigue (red lines/arrows indicate downregulation, blue lines/arrows indicate upregulation).

## Data Availability

No new data were created or analyzed in this study. Data sharing is not applicable to this article.

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
