# Peer review of "Molecular Mechanisms of Muscle Fatigue"

_ijms, 2021, doi:10.3390/ijms222111587_

Round 1

Reviewer 1 Report

This is a very nice review on the aspect of muscle fatigue, discussing MF in the various interdisciplinary contexts of overload, chronic disease and atrophy. I have only a few minor suggestions.

Abstract: "MF...it is intrinsically linked to the..."

Perhaps it is better to differentiate here between healthy M, taht might even induce a trophic response, and disease associated MF, such as by revision to " MF in disease conditions is...."

Top page 3, end of section 1, muscle fiber type distribution.  I think it would be helpful for the general reader to expand here somewhat on fiber types by mention a few skeletal muscles that are preferentially composed of fast fibers, or more rich in slow fibers, that different myosin heavy chain isoforms exist that are used for isoform fiber-typing, that the composition can be changed by training protocols somewhat, and citing perhaps work from Schiaffino on this.

Section 2, line 2:  "In the case of MS,...".  Is MS for multiple sclerosis? Of for MF (i.e. a typo?)

Author Response

Reviewer 1.

Comments and Suggestions for Authors

This is a very nice review on the aspect of muscle fatigue, discussing MF in the various interdisciplinary contexts of overload, chronic disease and atrophy. I have only a few minor suggestions.

Thank the reviewer for his/her helpful comments.

Q: Abstract: "MF...it is intrinsically linked to the...". Perhaps it is better to differentiate here between healthy M, taht might even induce a trophic response, and disease associated MF, such as by revision to " MF in disease conditions is...."

A: We have revised the text accordingly, which reads now: “…., the MF in illness is intrinsically…”

Q: Top page 3, end of section 1, muscle fiber type distribution.  I think it would be helpful for the general reader to expand here somewhat on fiber types by mention a few skeletal muscles that are preferentially composed of fast fibers, or more rich in slow fibers, that different myosin heavy chain isoforms exist that are used for isoform fiber-typing, that the composition can be changed by training protocols somewhat, and citing perhaps work from Schiaffino on this.

A: Two more paragraphs have been added to the text on page 3 to reflect the reviewer’s suggestions.

“The classification of muscle fibre types can be achieved by probing fibre type-specific molecular markers, such as myosin heavy chain isoforms. Typically, probing can be done through histochemical measurement of the myosin ATPase activity (low or high) or immunochemical identification with antibodies against myosin heavy chains {Schiaffino, 2018 #207}.”

“Most young and early adult humans in the general population have close to an even distribution of these two muscle fibre types in the muscles used for movement. However, soleus muscle located on the back of the lower limbs and posterior back muscles involved in maintaining posture contain mainly slow-twitch muscle fibres. Conversely, the vastus lateralis, which is the primary muscle of the thigh, contains fast-twitch fibres predominantly. A further departure from the normal muscle fibre distribution can be found in the muscles of marathoners and elite cyclists, where a high percentage of slow-twitch muscle fibres can be identified. Contrarywise this, the muscles of weightlifters and sprint runners show a high portion of fast-twitch muscle fibres. Although the ability of fibre types to shift from slow to fast and the other way around has been an ongoing subject of controversy, recent evidence may add more weight to the likelihood that such an event could occur with specific training regimens {Plotkin, 2021 #206}. However, these findings can be easily concealed by the existence of significant inter-individual responses {Schiaffino, 2021 #208}.”

Q: Section 2, line 2:  "In the case of MS,...".  Is MS for multiple sclerosis? Of for MF (i.e. a typo?)

A: Yes, we referred to multiple sclerosis at this location. To avoid confusion between the acronyms MF and MS, we have spelt out the full name of multiple sclerosis throughout the text.

Reviewer 2 Report

In the review article entitled “Molecular Mechanisms of Muscle Fatigue”, the authors summarized recent studies in the molecular mechanisms of muscle wasting, and explained the relationship between muscle wasting and muscle fatigue.

The introduction section offered basic knowledge of muscle architecture, which is very useful for the researchers in the different fields. Currently, the description of the molecular pathways may be familiar to all researchers in any field around the world. However, it can be expected that many of them lack the knowledge of muscle architecture. The introduction may kindly guide the readers to the muscle world.

I can also say that the authors provided a balanced view of the topic.

However, I think it would be better for the authors to discuss the relationship between muscle wasting and muscle fatigue in more detail. And, if possible, I would like to know how the authors think about the non-local muscle fatigue, which may be beyond the scope of their review article.

This is a very small thing, but I found that “MS” appeared at line 5 in page 3 of 16, but “multiple sclerosis” at line 25 in the same page (page 3 of 16), and “multiple sclerosis (MS) at line 4 in page 10 of 16.

Another very small thing. In page 3 of 16, “in cardiovascular and respiratory (COPD) disorders” could be revised to “in cardiovascular and respiratory disorders (e.g., heart failure, chronic obstructive pulmonary disease (COPD)). This revision is based on the abstract.

Author Response

Reviewer 2.

Comments and Suggestions for Authors

We thank the reviewer for his/her helpful comments.

In the review article entitled “Molecular Mechanisms of Muscle Fatigue”, the authors summarized recent studies in the molecular mechanisms of muscle wasting, and explained the relationship between muscle wasting and muscle fatigue.

The introduction section offered basic knowledge of muscle architecture, which is very useful for the researchers in the different fields. Currently, the description of the molecular pathways may be familiar to all researchers in any field around the world. However, it can be expected that many of them lack the knowledge of muscle architecture. The introduction may kindly guide the readers to the muscle world.

I can also say that the authors provided a balanced view of the topic.

Q: However, I think it would be better for the authors to discuss the relationship between muscle wasting and muscle fatigue in more detail.

A: We agree with the reviewer that it should have been more scope for a more detailed discussion around the relationship between muscle wasting and muscle fatigue, but because of space limitations, we could not presently expend as suggested. An expansion of this relationship should have also required additional literature describing outcomes of various interventions like resistive exercise, diet provisions with protein-rich formulations to improve muscle mass and muscle function, and thereby revoking muscle fatigue.

Additionally, such expansion would have diluted the message of the present review, which, as the title implies, was to summarise literature-based evidence of the molecular mechanisms underpinning muscle fatigue following initiation of muscle atrophy in chronic and acute illness. There is also, unfortunately, a scarce body of proof concerning interventional studies.

Q: And, if possible, I would like to know how the authors think about the non-local muscle fatigue, which may be beyond the scope of their review article.

A: This is an interesting question. Non-local muscle fatigue (NLMF) describes the functional impairment of a contralateral or remote non-exercised muscle following a strenuous exercise of a different muscle. We have found evidence that acute trauma (i.e., major surgery), in addition to exercise, can induce/initiate atrophy in a remote muscle. In the two works cited below, we provided molecular and biochemical evidence underpinning the documented atrophy and NLMF in a distal to trauma muscle.

“Inflammation-mediated muscle metabolic dysregulation local and remote to the site of major abdominal surgery. Clin Nutr. 2018; 37:2178–2185. “

“Major elective abdominal surgery acutely impairs lower limb muscle pyruvate dehydrogenase complex activity and mitochondrial function. Clin Nutr 2021 Mar; 40(3):1046-1051.”

This is a very small thing, but I found that “MS” appeared at line 5 in page 3 of 16, but “multiple sclerosis” at line 25 in the same page (page 3 of 16), and “multiple sclerosis (MS) at line 4 in page 10 of 16.

A: Yes, we meant multiple sclerosis at this location. To avoid confusion between the acronyms MF and MS, we have spelt out the full name of multiple sclerosis throughout the text.

Q: Another very small thing. In page 3 of 16, “in cardiovascular and respiratory (COPD) disorders” could be revised to “in cardiovascular and respiratory disorders (e.g., heart failure, chronic obstructive pulmonary disease (COPD)). This revision is based on the abstract.

A: We have revised the abstract text and on page 4 according to the reviewer’s suggestion.

Reviewer 3 Report

A clear, well written review that presents no major flaws. This review can be of tutorial importance. Some minor errors should be corrected as follows.

1. Acronysms should be specified as they appear the first time.

2. Chapter 5, page 5. (Figure 1) should be (Figure 2).

3. Chapter 6, page 5. (Figure 1) should be (Figure 2).

4. Chapter 12, page 6. (Figure 1) should be (Figure 2).

Author Response

Reviewer 3.

We thank the reviewer for his/her helpful comments.

Comments and Suggestions for Authors

A clear, well written review that presents no major flaws. This review can be of tutorial importance. Some minor errors should be corrected as follows.

Q1. Acronysms should be specified as they appear the first time.

Now we provide the full description of each acronym at the first appearance in the text

Q2. Chapter 5, page 5. (Figure 1) should be (Figure 2).

Changed to Figure 2.

Q3. Chapter 6, page 5. (Figure 1) should be (Figure 2).

Changed to Figure 2.

Q4. Chapter 12, page 6. (Figure 1) should be (Figure 2).

Changed to Figure 2.